# Antiviral Therapy for the Next Influenza Pandemic

**DOI:** 10.3390/tropicalmed4020067

**Published:** 2019-04-18

**Authors:** Aeron C. Hurt

**Affiliations:** 1WHO Collaborating Centre for Reference and Research on Influenza, VIDRL, Peter Doherty Institute for Infection and Immunity, Melbourne, VIC 3000, Australia; Aeron.Hurt@influenzacentre.org; Tel.: +613-9342-9314; 2Department of Microbiology and Immunology, University of Melbourne, Parkville, VIC 3010, Australia

**Keywords:** influenza, antivirals, pandemic, stockpile

## Abstract

Influenza antivirals will play a critical role in the treatment of outpatients and hospitalised patients in the next pandemic. In the past decade, a number of new influenza antivirals have been licensed for seasonal influenza, which can now be considered for inclusion into antiviral stockpiles held by the World Health Organization (WHO) and individual countries. However, data gaps remain regarding the effectiveness of new and existing antivirals in severely ill patients, and regarding which monotherapy or combinations of antivirals may yield the greatest improvement in outcomes. Regardless of the drug being used, influenza antivirals are most effective when treatment is initiated early in the course of infection, and therefore in a pandemic, effective strategies which enable rapid diagnosis and prompt delivery will yield the greatest benefits.

## 1. Commentary

The devastating influenza pandemic of 1918–1919 that resulted in an estimated 50 million deaths occurred in an era prior to modern vaccines, antivirals, antibiotics and advanced medical care that many of us now take for granted. Whilst a crude “mixed bacterial” vaccine was administered to approximately 8% of the Australian population during the pandemic in an effort to provide protection from infection (described in detail by G.D Shanks in this Special Issue) [1], the only therapeutic option available (to a small number of infected patients) was transfusion with influenza-convalescent human sera [2]. The treatment of patients during the 1918 pandemic represented some of the earliest use of convalescent sera. This was shown to reduce mortality from 37% in hospitalised untreated controls to 16% amongst treated patients, with further improvement if treatment was initiated early in their course of illness [2]. While the use of convalescent sera or pooled intravenous immunoglobulin from recovered patients remains a potential intervention in seriously ill patients with influenza in a future pandemic [3], the focus for modern influenza therapeutics has shifted to small-molecule compounds.

## 2. What Has Changed in the last Century When It Comes to the Treatment of Influenza? 

One hundred years on from the 1918 pandemic, we now have in our armoury a number of small-molecule compounds that target different parts of the influenza virus replication cycle [4]. The first antiviral that was licensed in the 1960s for the treatment and prophylaxis of influenza was amantadine, a compound from the adamantane class of drugs. Amantadine interferes with the M2 ion channel protein of influenza A viruses, thereby preventing proton transport and the initiation of viral replication [5]. However, the use of these antivirals has been limited in the past decade due to the emergence and spread of resistance amongst circulating strains [6]. Although a future pandemic strain may be adamantane-sensitive, this class of drugs has a high propensity to select for resistant viruses in treated patients [7], and therefore would not be the primary antiviral drug of choice for use in a pandemic.

The first rationally designed influenza antivirals were the neuraminidase inhibitors (NAIs) oseltamivir and zanamivir, which became available in 1999–2000. It was shown in randomised controlled trials (RCTs) that treatment with NAIs reduced symptom duration by approximately 24 h in otherwise healthy influenza patients [8]. Oseltamivir became the market leader due to its oral delivery being preferable over inhaled zanamivir, and whilst its use in treating uncomplicated seasonal influenza became commonplace in Japan (and to some degree the USA), most developed countries primarily used the NAIs for the treatment of hospitalised or severely ill patients. This has included the treatment of severely ill patients infected with A(H5N1) avian influenza viruses [9]. 

In 2018 a new influenza antiviral, baloxavir marboxil, was licensed in both Japan and the US. Baloxavir targets a different site of the influenza virus to that of the NAIs, as it inhibits the endonuclease of the viral polymerase complex of influenza A and B viruses. Although this compound had a similar effect on reducing symptom duration to that seen for oseltamivir, it had a significantly greater effect on reducing viral replication and shedding than oseltamivir treatment [10]. Data are not yet available on whether this enhanced reduction in viral replication will translate into reduced severity of disease in high-risk patients or whether it may also reduce secondary transmission. Viruses with reduced baloxavir susceptibility have been detected in treated patients and some untreated close contacts, with the highest frequencies being observed in children infected with A(H3N2) viruses [11,12].

## 3. Development of Antiviral Stockpiles for Pandemic Preparedness

In 2003, the world became aware of human cases of H5N1 avian influenza infection in Asia which were characterised by severe pneumonia and high mortality [9]. Concerns were immediately raised that this virus may cause the next pandemic. As a result, governments began developing pandemic preparedness plans that detailed how a country might respond should the virus become readily transmissible between humans. Because strain-specific influenza vaccines require at least four to six months to produce, antivirals became the most appropriate pharmaceutical control measure that could be mobilised immediately at the outset of a pandemic, given the broad activity of the compounds against all influenza A subtypes of human or animal origin. As such, many governments around the world began stockpiling oseltamivir (and to a lesser degree zanamivir) as part of their pandemic preparedness plans to aid in both the treatment of severely ill patients and for prophylaxis of front-line heath and emergency services workers. Although this was seen as a positive move by many organisations (e.g. the World Health Organization), the purchase of antiviral stockpiles was criticised by some because the evidence for NAI effectiveness in severely ill or hospitalised patients relied on observational studies (which are subject to uncontrolled bias) rather than RCTs [13]. The lack of RCT data in hospitalised patients is in part due the ethical constraints of conducting a placebo-controlled study in a group of patients where the standard of care (SOC) now includes oseltamivir. However, as new antivirals such as baloxavir become licensed for use in otherwise healthy individuals, it becomes particularly important to evaluate whether they provide additional benefit to SOC in a robust RCT. Nevertheless, evidence derived from observational studies of serious outcomes consistently suggests that oseltamivir reduces the risk of death by approximately half if treatment is initiated within 48 hours of symptom onset [14,15]. As such, at this point in time oseltamivir remains an important component of many countries’ stockpiles.

## 4. Access to Antiviral Stockpiles in the 2009 Pandemic

In 2009, the world experienced an influenza pandemic caused not by H5N1 or another avian influenza virus, but instead by an H1N1 virus that was derived from viruses circulating in pigs. This was the first pandemic to occur in an era of influenza antivirals and at a time where many developed countries had stockpiles on hand. However, the quantity of antivirals used and the speed with which they were made available differed substantially across different countries. In Japan, NAI use for seasonal influenza was widespread for nearly a decade prior to 2009. This likely led to the rapid delivery and extensive use of the drugs in the pandemic. Over 98% of hospitalised children in Japan with pandemic influenza virus infections were treated with an NAI, with 89% receiving the antivirals within 48 h and 70% within 24 h [16]. Only 1% of the hospitalised children ultimately required mechanical ventilation, and only one death was recorded [16]. Amongst pregnant women in Japan with pandemic influenza, >90% were given NAIs within 48 h of symptom onset, and many were treated prophylactically after close contact with an infected person. Compared to the high global mortality rates seen in pregnant women [17], Japan reported no deaths caused by influenza in this group of patients [18]. A meta-analysis of global data from the 2009 pandemic also showed that delivery of antiviral treatment within 48 hours of symptom onset reduced the likelihood of hospitalisation of patients with comorbidities [19]. In countries where routine (seasonal) NAI usage was lower, novel strategies were implemented to try and achieve rapid access to antivirals. In the UK, individuals with influenza-like illness could access a national telephone hotline where, through a series of clinical questions, it was evaluated if antiviral treatment was appropriate and whether a medical General Practitioner (GP) consultation was necessary. Antivirals could then be collected at various “antiviral collection points” by a non-infected friend or family member. However, despite this initiative, many in the high-risk groups, such as pregnant women, failed to access these medications early and prior to hospital admission [20]. In other countries, where antivirals were delivered via GPs or hospitals, significant delays were observed when stockpiles were centralised [21]. Decentralising stockpiles into local centres, such as major hospitals, would not only facilitate more rapid treatment of ill patients during a pandemic; it would also allow the periodic use of the stockpile for the treatment of inter-pandemic seasonal influenza to avoid wastage due to “shelf-life” expiration [21]. However, this strategy would require the stockpile to be regularly “topped-up” to ensure that sufficient stock remained for a pandemic response.

## 5. In the Future, What Might Antiviral Stockpiles Look like and How Might They Be Used? 

With the recent licensure of baloxavir, the first of the new class of polymerase inhibitors, there are opportunities to diversify antiviral stockpiles, beyond just NAIs (Table 1). Having multiple classes/types of antiviral with different modes of action may prove beneficial should a pandemic virus develop resistance. In addition to baloxavir, pimodivir and favipiravir are polymerase inhibitors which may play a role in antiviral stockpiles in the future. Pimodivir is currently undergoing phase III trials, and if licensed for seasonal influenza would also be considered for pandemic use. Favipiravir is already part of the Japanese antiviral stockpile. However, because of concerns around its teratogenicity, the Japanese Government will only allow its use if a pandemic virus has developed resistance to other available compounds, and it presumably would not be used for the treatment of pregnant women. Having multiple antiviral agents on hand also provides opportunities for delivering the compounds as combinations. This approach has the potential for enhanced effectiveness and may also decrease the risk of resistant variants arising, although combination therapy will of course be more costly than monotherapy. A study to assess the efficacy of baloxavir together with SOC (which typically involves NAI treatment) in hospitalised patients with severe influenza is currently ongoing (NCT03684044; clinicaltirals.gov), and will provide very important information on the benefit of combination therapy in this group of patients. 

Modelling studies have evaluated different strategies for the use of antivirals in a future pandemic. It is generally considered that containment strategies, via pre-exposure antiviral prophylaxis, will consume large quantities of the stockpile. Although this approach has been successful in closed settings (e.g., a Singaporean military camp) [22], it is unlikely to contain the spread when the virus is circulating more widely. NAI treatment strategies that allow the liberal distribution of antivirals for early treatment in outpatient settings are likely to result in the greatest reductions in hospitalisations, critical care interventions and deaths. Restricting community-based treatment to high-risk groups, while effective in those groups, is unlikely to prevent large numbers of cases arising from lower-risk individuals who comprise the majority of the population [23].

Based on the Japanese example, it is clear that an existing level of familiarity with antiviral prescribing amongst clinicians for the management of seasonal influenza is an important enabler in ensuring a timely and wider response during a pandemic. Influenza antivirals, regardless of the compound being used, are more effective when delivered early in the course of illness. Therefore, countries that have limited influenza antiviral usage for seasonal influenza need to strongly consider rapid distribution and access strategies for a pandemic. These include phone hotlines and over-the-counter availability of the drug in pharmacies without the need for a prescription (as is used in New Zealand), rather than relying on traditional routes such as GP and hospital outpatient visits. The expense of purchasing and maintaining influenza antiviral stockpiles also means that many low-income countries will remain dependent upon stockpiles held by the World Health Organization for dealing with severe seasonal outbreaks or a pandemic. 

## 6. Conclusions

Since the 2009 influenza pandemic there has been a significant increase in the number of influenza antivirals that have either been approved for use or that have advanced through the clinical trial pipeline. This has opened up the potential to radically diversify stockpiles in preparation for future pandemics. However, it is important to improve our understanding of the effectiveness and limitations of these compounds in treating severely ill or hospitalised patients, as well as which antiviral or combinations of antivirals may yield the greatest improvement in outcomes. Regardless of the drug that is used, influenza antivirals are most effective when treatment is initiated early in the course of infection. Therefore, it is important for countries to establish protocols for rapid diagnosis and timely access to antivirals during a pandemic to ensure that the greatest benefits of an antiviral stockpile are realised.

## Figures and Tables

**Table 1 tropicalmed-04-00067-t001:** Overview of influenza antivirals used in the 2009 pandemic and those now licensed and available for use in future pandemics.

Antiviral (Trade Name)	Mode of Action	Use in the 2009 Pandemic	Potential Use in a Future Pandemic
Amantadine/rimantadine (Symmetrel/Flumadine)	M2 ion channel inhibitor	Limited/no use due to the 2009 pandemic virus being adamantane-resistant at the time of emergence	Unlikely to be used in a future pandemic due to rapid selection of resistance
Oseltamivir (Tamiflu)	Neuraminidase inhibitor	Major component of WHO and country stockpiles	Likely to remain a part of future stockpiles due to long “shelf-life”, ease of oral delivery and familiarity with its use for seasonal influenza
Zanamivir (Relenza)	Neuraminidase inhibitor	Minor component of WHO and some country stockpiles	Likely to make up only a minor component or not be used due to inhaled delivery and low use for seasonal influenza. Has a low propensity to select for resistance, which is a benefit
Peramivir (Rapivab)	Neuraminidase inhibitor	Some use in Japan where it was licensed. Small usage elsewhere under emergency use authorisation only	May be utilised in small quantities given it is approved for intravenous delivery, which may be optimal for some severely ill patients
Laninamivir (Inavir)	Neuraminidase inhibitor	Was not available	May be used in Japan (the only country to license the antiviral for seasonal influenza use). Has benefits of single dose and low propensity to select for resistance, but is delivered via inhalation
Favipiravir (Avigan)	Polymerase inhibitor (purine nucleoside altering role of PB1)	Was not available	Limited use due to concerns of teratogenicity. May be used in Japan if pandemic virus is resistant to other available antivirals. Unlikely to be used elsewhere
Baloxavir (Xofluza)	Polymerase inhibitor (PA endonuclease)	Was not available	Likely to be a part of future stockpiles due to ease of dosing and delivery (single oral dose) and rapid virological effect

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
