# Peer review of "Antiviral Therapy for the Next Influenza Pandemic"

_tropicalmed, 2019, doi:10.3390/tropicalmed4020067_

Round 1

Reviewer 1 Report

This is an excellent commentary addressing the issues of stockpiling of antivirals to prepare for the potential influenza outbreaks.  The author considered the properties of the current available antivirals and the pros and cons of each of them, including the clinical experience (particularly Japan experience), drug resistance and other practical factors, such as shelf life of drugs, etc.  This is a timely and well-informed commentary.

1)   Drug resistance to beloxavir has recently emerged, particularly in young children in Japan.  This is an important issue relevant to this topic.  This should be included in this commentary to make it timely.

2)   Cost of each antiviral and the potential financial burden caused by these drugs should be considered.

3)   Line 93: “no death of” should be changed to “no death”

Author Response

Reviewer #1

This is an excellent commentary addressing the issues of stockpiling of antivirals to prepare for the potential influenza outbreaks.  The author considered the properties of the current available antivirals and the pros and cons of each of them, including the clinical experience (particularly Japan experience), drug resistance and other practical factors, such as shelf life of drugs, etc.  This is a timely and well-informed commentary.

1)   Drug resistance to baloxavir has recently emerged, particularly in young children in Japan.  This is an important issue relevant to this topic.  This should be included in this commentary to make it timely.

RESPONSE: A line addressing this was already present at the end of the “What has changed in the last century when it comes to the treatment of influenza?” paragraph. However I have now added an additional recent citation for a paper published a week ago, and have added a few additional words to mention that viruses with reduced baloxavir susceptibility have also been found in untreated contacts of those shedding resistant viruses. It now reads: 

“Viruses with reduced baloxavir susceptibility have been detected in treated patients and some untreated close contacts, with the highest frequencies being observed in children infected with A(H3N2) viruses 11,12.”

2)   Cost of each antiviral and the potential financial burden caused by these drugs should be considered.

RESPONSE: The cost of each antiviral to the consumer from a pharmacy is not only difficult to find, but differs substantially between countries, and has little relevance to the costs of purchasing these drugs in large volumes for a pandemic stockpile. For this reason this information has not been included. However, it is important to make mention that there is a substantial cost involved in purchasing an antiviral stockpile and therefore the following sentence has been added:  “The expense of purchasing and maintaining influenza antiviral stockpiles also means that many low-income countries will remain dependent upon stockpiles held by the World Health Organization for dealing with severe seasonal outbreaks or a pandemic. “ 

3)   Line 93: “no death of” should be changed to “no death”

RESPONSE: This has been modified.

Reviewer 2 Report

The manuscript describes the approved antivirals on market for treating influenza A infections. The review is well written and concise, but can be improved with the following suggestions:

a)    The role of antivirals during pandemic in high risk population as separate section would be helpful.  In addition to pregnant mothers, the outcome of antivirals in children, elderly and other comorbidities can provide some insights how the efficacy and outcome flu antivirals during pandemic. 

b)   A few sentences on the clinical outcome of antivirals during 2009 pandemic in Europe, Africa and Americas would be strengthen the review.

c)    Conclusion needs to be bold. Some key points ideas can be stressed. Example: how to improve the effectiveness of antivirals during future pandemics? We already know that strain diversity of H3N2 created enormous morbidity in several countries during last 2 seasons. Are there any novel compounds in the Pipeline? What combinations of antivirals will work best based on our current knowledge. 

d) Baloxavir was approved in 2018 and readers may not be familiar how this drug was developed. Give the reader a sense of optimism. The fact that this is the first target that targets a region of the virus that may not be as prone to mutations like HA and NA points out future targets that directly affects virus replication specially for flu which is so unique.

Author Response

The manuscript describes the approved antivirals on market for treating influenza A infections. The review is well written and concise, but can be improved with the following suggestions:

a)    The role of antivirals during pandemic in high risk population as separate section would be helpful.  In addition to pregnant mothers, the outcome of antivirals in children, elderly and other comorbidities can provide some insights how the efficacy and outcome flu antivirals during pandemic. 

RESPONSE: An additional sentence describing the benefit of antiviral treatment in high risk patients has now been added. This section now includes data for pregnant women, children and those with co-morbidities.  The current section now reads:

“Over 98% of hospitalized children in Japan with pandemic influenza virus infections were treated with an NAI, with 89% receiving the antivirals within 48 hours and 70% within 24 hours 16. Only 1% of the hospitalized children ultimately required mechanical ventilation, and only one death was recorded 16. Amongst pregnant women in Japan with pandemic influenza, >90% were given NAIs within 48 hours of symptom onset, and many were treated prophylactically after close contact with an infected person. Compared to the high global mortality rates seen in pregnant women 17, Japan reported no deaths caused by influenza in this group of patients 18. A meta-analyses of global data from the 2009 pandemic also showed that delivery of antiviral treatment within 48 hours of symptom onset reduced the likelihood of hospitalization of patients with co-morbidities 19.”

b)   A few sentences on the clinical outcome of antivirals during 2009 pandemic in Europe, Africa and Americas would be strengthen the review.

RESPONSE:  The best effectiveness data comes from meta-analyses of global 2009 pandemic data rather than data from individual countries or regions as they have large amounts of data that can be adjusted for potential biases. These global meta-analyses are already described and cited in the review (e.g. Muthuri et al; Venkatesan et al). The author doesn’t feel that citing other studies from other regions will add substantially to the text already provided.

c)    Conclusion needs to be bold. Some key points ideas can be stressed. Example: how to improve the effectiveness of antivirals during future pandemics? We already know that strain diversity of H3N2 created enormous morbidity in several countries during last 2 seasons. Are there any novel compounds in the Pipeline? What combinations of antivirals will work best based on our current knowledge. 

RESPONSE: These key points are already covered in the section “In the future, what might antiviral stockpiles look like and how might they be used?”. The author feels that the text pasted below describes the new compounds that are in the pipeline, the concept of combination therapy, and the data that will arise in the next year or so that will inform on the most relevant combination (oseltamivir and baloxavir). Going beyond the compounds mentioned in the review, to include those in phase I and II trials, is of limited value as they will be someway off being used for pandemic stockpiling. The current text that reflects the reviewers request  

“In addition to baloxavir, pimodivir and favipiravir are polymerase inhibitors which may play a role in antiviral stockpiles in the future. Pimodivir is currently undergoing phase III trials, and if licensed for seasonal influenza would also be considered for pandemic use. Favipiravir is already part of the Japanese antiviral stockpile, but because of concerns around teratogenicity, the Japanese Government will only allow its use if a pandemic virus has developed resistance to other available compounds, and presumably would not be used for treatment of pregnant women. Having multiple antiviral agents on hand also provides opportunities for delivering the compounds as combinations. This approach has potential for enhanced effectiveness and may also decrease the risk that resistant variants arise, although combination therapy will of course be more costly than monotherapy. A study to assess the efficacy of baloxavir together with SOC (which typically involves NAI treatment) in hospitalised patients with severe influenza is currently ongoing (NCT03684044; clinicaltirals.gov), and will provide very important information on the benefit of combination therapy in this group of patients. “

d) Baloxavir was approved in 2018 and readers may not be familiar how this drug was developed. Give the reader a sense of optimism. The fact that this is the first target that targets a region of the virus that may not be as prone to mutations like HA and NA points out future targets that directly affects virus replication specially for flu which is so unique.

RESPONSE: some additional text has now been added to the section “What has changed in the last century when it comes to the treatment of influenza?” This section now reads:

“In 2018 a new influenza antiviral, baloxavir marboxil, was licensed in both Japan and the US. Baloxavir targets a different site of the influenza virus to that of the NAIs, as it inhibits the endonuclease of the viral polymerase complex of influenza A and B viruses.”

Round 2

Reviewer 2 Report

None